# Remdesivir: Effectiveness and Safety in Hospitalized COVID-19 Patients—Analysis of Retrospectively Collected Data from Daily Practice in the Omicron Variant Era and Comparison with the Pre-Omicron Period

**DOI:** 10.3390/microorganisms13102242

**Published:** 2025-09-25

**Authors:** Nikos Pantazis, Evmorfia Pechlivanidou, Vassiliki Rapti, Dimitra Kavatha, Haralampos Milionis, Ioannis Kalomenidis, Karolina Akinosoglou, Periklis Panagopoulos, Symeon Metallidis, Diamantis Kofteridis, Nikolaos V. Sipsas, Ioannis Katsarolis, Garyfallia Poulakou, Anastasia Antoniadou, Eirini Christaki, Theofani Rimpa, Markos Marangos, Vasileios Petrakis, Olga Tsachouridou, Vasiliki E. Georgakopoulou, Pinelopi Kazakou, Sotirios P. Karagiannis, Eleni Polyzou, Giota Touloumi

**Affiliations:** 1Department of Hygiene, Epidemiology and Medical Statistics, Medical School, National & Kapodistrian University of Athens, 11527 Athens, Greece; pevmorfia@med.uoa.gr (E.P.); gtouloum@med.uoa.gr (G.T.); 23rd Department of Internal Medicine, Athens Hospital for Diseases of the Chest “Sotiria”, Medical School, National & Kapodistrian University of Athens, 11527 Athens, Greece; vrapt@med.uoa.gr (V.R.); gpoulakou@gmail.com (G.P.); 34th Department of Internal Medicine, Attikon University General Hospital, Medical School, National & Kapodistrian University of Athens, 12462 Athens, Greece; dimitra.kavatha@gmail.com (D.K.); ananto@med.uoa.gr (A.A.); pkazakou@hotmail.com (P.K.); 4Department of Internal Medicine, University General Hospital of Ioannina, Faculty of Medicine, School of Health Sciences, University of Ioannina, 45500 Ioannina, Greece; hmilioni@uoi.gr (H.M.); eirini.christaki@uoi.gr (E.C.); 51st Department of Critical Care & Pulmonary Service, Evangelismos General Hospital, Medical School, National & Kapodistrian University of Athens, 10676 Athens, Greece; ikalom@med.uoa.gr (I.K.); fanhriba@gmail.com (T.R.); sotiriskaragiann@gmail.com (S.P.K.); 6Department of Internal Medicine, University General Hospital of Patras, Department of Medicine, University of Patras, 26504 Rio, Greece; akin@upatras.gr (K.A.); marangos@upatras.gr (M.M.); polyzou.el@gmail.com (E.P.); 7Department of Internal Medicine, University General Hospital of Alexandroupolis, Department of Medicine, Democritus University of Thrace, 68100 Alexandroupolis, Greece; ppanago@med.duth.gr (P.P.); vasilispetrakis1994@gmail.com (V.P.); 81st Department of Internal Medicine, AHEPA University General Hospital, Medical School, Aristotle University of Thessaloniki, 54636 Thessaloniki, Greece; metallidissimeon@yahoo.gr (S.M.); olgat_med@hotmail.com (O.T.); 9Department of Internal Medicine, University General Hospital of Herakleion, Faculty of Medicine, School of Health Sciences, University of Crete, 71110 Herakleion, Greece; 10Infectious Diseases and COVID-19 Unit, Laiko General Hospital, National & Kapodistrian University of Athens, 11527 Athens, Greecevaso_georgakopoulou@hotmail.com (V.E.G.); 11Medical Affairs, Gilead Sciences Hellas and Cyprus, 17564 Paleo Faliro, Greece; ioannis.katsarolis@gilead.com

**Keywords:** COVID-19, remdesivir, hospitalized patients, efficacy, safety, Pre-Omicron vs. Omicron, Omicron

## Abstract

Severe acute respiratory syndrome coronavirus-2 (SARS-CoV-2) has impacted global health. Remdesivir was approved based on clinical trials demonstrating improved outcomes in hospitalized patients. The ReEs-COVID19 study provides real-world evidence on its effectiveness and safety across two periods: Pre-Omicron and Omicron. This retrospective, observational cohort study included 1610 patients hospitalized with COVID-19, treated with remdesivir during Pre-Omicron (September 2020–February 2021; n = 606) and Omicron (June 2022–March 2023; n = 1004) periods. Primary endpoint: time to discharge; Hepatic/renal function abnormalities were also investigated. In the Omicron period patients were older and had more comorbidities but remdesivir was initiated earlier (median: 2 days from symptom onset) compared to the Pre-Omicron period (8 days). ICU admissions rates and direct COVID-19-related deaths were significantly lower, but overall 30-day mortality was higher during the Omicron period. Earlier remdesivir administration was associated with faster discharge. Abnormal liver tests and acute kidney injury were rare across both periods. ReEs-COVID19 confirmed remdesivir’s effectiveness and safety in real-world clinical settings during both periods, underscoring its importance in treatment of hospitalized COVID-19 patients, especially when initiated earlier in the disease course. Further research is needed to evaluate its utility in specific subgroups (e.g., immuno-compromised) and in combination with other treatments.

## 1. Introduction

Severe acute respiratory syndrome coronavirus 2 (SARS-CoV-2), the causative agent of Coronavirus disease 2019 (COVID-19), was initially found in Wuhan, China, in late 2019 [1]. In response to the virus’s swift proliferation, the World Health Organization (WHO) proclaimed a pandemic on 12 March 2020 [1]. Globally, over 776 million individuals have been diagnosed with COVID-19, resulting in 7.1 million deaths [1]. The scientific response to COVID-19 was extraordinary, with extensive research on viral biology, diagnostics, clinical infection features, prevention strategies, and the development of vaccines and treatment alternatives [2,3].

Remdesivir (Veklury^®^, Gilead Sciences, Foster City, CA, USA) (GS-5734) was identified early in 2020 as a potential treatment option for COVID-19 because of its capacity to block SARS-CoV-2 in vitro [4,5,6]. Following the initial promising results [4,7,8,9], numerous clinical trials have examined its efficacy and safety. The findings of ACTT-1, demonstrating that the median time to recovery was reduced by 5 days in the remdesivir group relative to the placebo group [10], along with two additional clinical trials (GS-US-540-5773, GS-US-540-5774) [10,11,12], led to the conditional approval of remdesivir on 3 July 2020 by the European Medicines Agency (EMA) and full approval on 22 October 2020 by the US Food and Drug Administration (FDA) [13,14,15].

Remdesivir was initially administered in Greece as a part of the ACTT-1 clinical trial in 4 hospitals or as part of a compassionate use program. Following its EMA original approval, remdesivir became readily available from September 2022, was included in national guidelines and was widely used in COVID-19 clinics around the country [16].

Due to the inherently high mutation rate of RNA viruses, SARS-CoV-2 undergoes continuous adaptive evolution [17]. This has led to the emergence of genetic variants that the WHO designated as variants of concern (VOCs) and which, after comparative evaluation, have been found to differ in terms of transmissibility and virulence while the corresponding efficacy of vaccines and therapeutics against them was also varying [18].

In Europe, three major variants contributed to the COVID-19 pandemic: Alpha, which was active in the first quarter of 2020, Pre-Omicron, which dominated from June to December 2021, and Omicron, which has been the dominant lineage since early 2022 and continues to evolve with many subvariants [19].

Remdesivir has demonstrated efficacy against multiple SARS-CoV-2 variants due to its mechanism of targeting RNA-dependent RNA polymerase (RdRp), rather than the spike protein [20,21]. However, mutations in RdRp, such as P314L and P323L, have emerged over time, potentially impacting remdesivir ‘s effectiveness [22,23,24], while S759A and V792I mutations were highlighted by experimental studies as potential blockers of remdesivir’s activity [25]. Despite these challenges, clinical trials and real-world data confirm remdesivir’s effectiveness during pandemic waves caused by earlier variants, including Pre-Omicron [1,10,11,12,15,26,27,28,29]. Although in vitro studies [30] indicate that remdesivir retains activity against the Omicron variant at a level similar to that observed with the Pre-Omicron variant, there is a paucity of confirmation in clinical settings [26,31]. Remdesivir’s unimpeded inclusion in treatment guidelines underscores its importance as the one and only effective antiviral agent against all waves of COVID-19.

The goal of the ReEs-COVID19 (Remdesivir Effectiveness and Safety in COVID-19) study was to enrich the knowledge obtained through remdesivir clinical trials by using real-world evidence from daily clinical practice, involving data from both Pre-Omicron and Omicron periods. We hypothesized that remdesivir would remain effective in improving clinical outcomes across different SARS-CoV-2 variant periods, including Pre-Omicron and Omicron, and that early administration would be associated with faster discharge. More specifically, ReEs-COVID19 aims to describe patterns and trends in remdesivir use during the Omicron period, evaluate effectiveness and safety when provided to hospitalized patients, and investigate relevant differences between the Omicron and Pre-Omicron eras.

## 2. Materials and Methods

### 2.1. Study Design and Eligibility Criteria

ReEs-COVID19 is a retrospective observational cohort study of adults (≥18 years old) who received remdesivir as part of their care while being hospitalized for PCR documented SARS-CoV-2 infection for at least 24 h in one of the 6 collaborating clinics. Pre-Omicron-variant era (1 September 2020 to 28 February 2021) corresponds to a period during which remdesivir was widely available in Greek hospitals, and it was just before the start of mass vaccination campaign in the country. Omicron-variant era (1 June 2022 to 31 March 2023) aligns with the period when Omicron emerged as the predominant variant in the country, and it follows the extensive mass vaccination campaign conducted nationwide. Patients admitted to the intensive care unit (ICU) on the initial day of hospitalization were excluded from both periods of the study. Similarly, those participating in clinical trials for alternative COVID-19 treatments, were also excluded to avoid overlapping interventions and potential confounding.

Data were gathered from patient files and hospital records. The information collected encompassed demographic and social characteristics, clinical status at the onset of hospitalization, medical history, risk factors and data regarding the course and outcomes of the hospitalization as well as any observations of elevated liver or kidney biochemical tests up to 30 days post-admission.

4C Mortality Score [32] at admission was calculated assuming that all participants had normal consciousness (Glasgow Coma Scale, GCS = 15) as GCS evaluations were not available. This assumption may slightly underestimate risk in patients with impaired consciousness, but the 4C score was used only for descriptive purposes.

Severity at baseline was defined using the ACTT-1 criteria [10] (severe disease: requiring invasive or non-invasive mechanical ventilation or requiring supplemental oxygen or SpO_2_ ≤ 94% on room air or tachypnea defined as respiratory rate ≥ 24 breaths per minute; mild/moderate disease: all other cases). Additionally, patients were classified into 3 groups based on their need for O_2_ at admission (1: not requiring O_2_, 2: requiring low flow O_2_, 3: requiring high flow O_2_ or invasive or non-invasive mechanical ventilation).

### 2.2. Endpoints

The main endpoint was the time from hospital admission to hospital discharge (with or without limitation of activities) within 30 days after hospital admission. Additionally, clinical status at 7, 14 and 30 days after admission was evaluated.

Patterns of remdesivir use were described through time between admission and 1st dose and total duration of administration. It should be noted that during both study periods, national guidelines suggested the standard remdesivir scheme of 200 mg intravenous on day 1, followed by 100 mg for the subsequent 4 days for patients in need of supplemental oxygen as per its label (excluding mechanically ventilated patients). The suggested timing for administration of the first dose was within 7 days since the positive PCR test and/or symptoms start [33].

Laboratory abnormalities in hepatic and renal markers were used to evaluate remdesivir’s safety profile. Abnormal liver function was classified as mild/moderate or severe according to aminotransferase (ALT) and/or aspartate aminotransferase (AST) level thresholds: (i) mild/moderate: ALT and/or AST between 3 and 5 times the upper limit of normal levels (ULN) and (ii) severe: ALT and/or AST above 5 × ULN. For renal function, the three-stage classification proposed by the Acute Kidney Injury (AKI) working group (KDIGO) was used [34,35]. Any related events were classified as probable remdesivir -associated adverse events if observed during remdesivir administration and up to 2 days after. The cut-off point of 2 days was set according to anticipated remdesivir pharmacokinetics [13,36].

### 2.3. Ethics

The Ethics and Deontology Committee of the Medical School of the National and Kapodistrian University of Athens along with the Scientific Council of each hospital participating in ReEs-COVID19 approved the study. The data were recorded anonymously through a random numeric code. In any case, the collection, transfer and analysis of data were based on the guidelines for good clinical practice and the relevant national regulations.

### 2.4. Statistical Analysis

Baseline characteristics were summarized using standard descriptive statistics. Categorical variables were compared using the Chi-square test (or Fisher’s exact test as appropriate), while continuous variables were compared using the Mann–Whitney U test. Time to discharge was analyzed using survival analysis techniques with censoring at 30 days. Data for patients who were not discharged or died within 30 days were censored at day 30. All logrank tests and Cox models, were stratified by baseline disease severity (ACTT-1 based definition) [10]. A *p*-value < 0.05 was considered statistically significant.

Statistical analysis was performed using Stata (StataCorp. 2023. Stata Statistical Software: Release 18. StataCorp LLC., College Station, TX, USA).

## 3. Results

### 3.1. Patients’ Characteristics

During the two study periods 1610 hospitalized for COVID-19 patients (Pre-Omicron n = 1004; Omicron n = 606) were included. Detailed demographic, clinical and laboratory characteristics of the patients at baseline according to variant period are presented in Table 1 (laboratory values by variant are given in Table A1).

Baseline disease severity differed significantly (*p* < 0.001) between the two periods with 74.2% of patients classified as having severe COVID-19 during the Pre-Omicron and 63.0% during the Omicron era. Similarly, more patients required supplementary O_2_ during the Pre-Omicron (60.8%) compared to the Omicron (54.3%) period (*p* = 0.001).

The proportion of male patients was higher during the Pre-Omicron period (60.6%) compared to the Omicron period (52.6%; *p* = 0.002). Patients in the Omicron era were older (median age: 76 years [IQR 66–85]) compared to the Pre-Omicron era (median age: 61 years [IQR 51–72.5]; *p* < 0.001). Patients hospitalized during the Omicron period were more likely to have two or more comorbidities (64.9% vs. 48.1%, *p* < 0.001), with a notable increase in the percentage of patients with active cancer (17.7% vs. 4.9%, *p* < 0.001). These differences were also reflected in the 4C Mortality Score for COVID-19 which was significantly (*p* < 0.001) higher for patients hospitalized during the Omicron compared to those in the Pre-Omicron period (median 10 vs. 7, respectively) and the Charlson Comorbidity Index (median 4 vs. 2, respectively). Among the 606 patients from the Omicron era, 470 (77.6%) had received at least one dose of a COVID-19 vaccine. Of these, 408 (86.8%) were fully vaccinated, defined as two primary doses plus a booster. The median (IQR) time since the last vaccination was 8.3 (6.1–11.4) months. The remaining 136 patients (22.4%) were unvaccinated. None of the pre-Omicron patients were vaccinated.

### 3.2. Remdesivir Patterns of Use

The median time from admission to remdesivir administration was significantly (*p* < 0.001) shorter during the Omicron period (0 days, IQR: 0–0) compared to the Pre-Omicron period (1 day, IQR: 1–2). Similarly, the median time between symptom onset and remdesivir administration was shorter during the Omicron period (2 days, IQR: 0–4) compared to the Pre-Omicron period (8 days, IQR: 5.5–10, *p* < 0.001). The median duration of remdesivir treatment was the same across both periods (5 days) with 77.7% of those hospitalized in the Pre-Omicron period and 79.7% of those in the Omicron receiving remdesivir for exactly 5 days. However, statistically significant differences (*p* < 0.001) were observed in treatment duration categories: fewer patients in the Pre-Omicron period received treatment for less than 5 days (89/1004 patients, 8.9%) compared to the Omicron period (83/606 patients, 13.7%, *p* < 0.001), and a lower percentage of patients in the Omicron period received treatment for 6–10 days (40/606 patients, 6.6%) compared to the Pre-Omicron period (132/1004 patients, 13.1%) (Figure 1). It is noteworthy that in the Omicron era, patients who received remdesivir treatment for 6–10 days, were more likely to have active cancer (55% vs. 15%), cardiovascular disease (78% vs. 57%) or requiring high flow O_2_/mechanical ventilation (33% vs. 10%) at baseline compared to those who received remdesivir for up to 5 days.

Regarding additional treatments, dexamethasone use was significantly less common during the Omicron period (170/606 patients, 28.1%) compared to the Pre-Omicron period (778/1004 patients, 77.5%, *p* < 0.001). Similarly, the use of anticoagulants was lower during the Omicron period (367/606 patients, 60.6%) than during the Pre-Omicron period (971/1004 patients, 96.7%, *p* < 0.001). In the Omicron period 41 (6.8%) patients received anakinra and 20 (3.3%) tocilizumab. The corresponding figures in the pre-Omicron period were 22 (2.2%) and 12 (1.2%).

### 3.3. Clinical Progression and Mortality

During the Omicron period, clinical status based on the 8-degree ordinal scale demonstrated significant improvement over time. At baseline, 70 (11.6%) required non-invasive ventilation or high-flow O_2_ (stage 6), 259 (42.7%) required low flow O_2_ (stage 5), and 277 (45.7%) were hospitalized but not requiring O_2_ while needing medical care (stage 4).

By day 7, 259 (42.7%) had been discharged with no activity limitations (stage 1), and 62 (10.2%) remained on non-invasive ventilation or high-flow oxygen (stage 6). At day 14, 438 (72.3%) were no longer hospitalized and had no limitations (stage 1), while 34 (5.6%) patients were discharged with limitations or requiring home oxygen (stage 2).

By day 30, 475 (78.4%) had no activity limitations (stage 1), and 48 (7.9%) patients were discharged but experienced limitations or required home oxygen (stage 2). The proportion of patients requiring hospitalization with oxygen support (stage 5) decreased from 259 (42.7%) at baseline to 16 (2.6%) by day 30. Similarly, those on non-invasive ventilation or high-flow oxygen (stage 6) decreased from 70 (11.6%) at baseline to just 1 patient (0.2%) at day 30 (Figure 2).

In the pre-Omicron period, at baseline, 8.6% of patients were on non-invasive ventilation or high-flow oxygen (stage 6), 52.2% required low-flow oxygen (stage 5), and 39.2% were hospitalized without oxygen but needed medical care (stage 4). By 30 days, 93.0% were not hospitalized and had no limitations (stage 1), while 1.7% remained hospitalized without oxygen (stage 4), 0.2% required oxygen (stage 5), 0.5% were on non-invasive ventilation (stage 6), 0.8% were on invasive mechanical ventilation or ECMO (stage 7), and 3.8% had died (stage 8).

ICU admission rates were significantly lower during the Omicron period, with only 8 patients (1.3%) requiring intensive care compared to 45 patients (4.5%) during the Pre-Omicron period (*p* < 0.001). A total of 55 patients (9.1%) had died by day 30 in the Omicron period. The corresponding mortality was significantly lower during the Pre-Omicron period (38 deaths, 3.8%; *p* < 0.001) but COVID-19-attributable deaths (as determined and recorded by the treating physicians) were more frequent in the Pre-Omicron period (31/38; 81.6% of deaths) compared to the Omicron one (5/55; 9.1%; *p* < 0.001). In univariable analysis, mortality appeared higher during the Omicron period (OR 2.54, 95% CI 1.66–3.89; Table A2). After adjustment for age, comorbidities, and baseline severity, this association was no longer significant (adjusted OR 1.23, 95% CI 0.76–1.97; Table A3). Within the Omicron period, vaccination was independently associated with reduced mortality (adjusted OR 0.45, 95% CI 0.25–0.81; Table A4). Clinical progression and discharge rates for the two periods are depicted in Figure A1.

The median age of the deceased during the Omicron period was 84 years (IQR: 76–92), with 35 patients (63.6%) aged 80 or older and 13 (23.6%) aged 70–79. Most patients (49, 89.1%) resided in urban areas, while 5 (9.1%) lived in rural areas, and 1 (1.8%) in a semi-urban area. The vast majority (54, 98.2%) were of Greek nationality. Baseline disease status, according to ACTT-1 study, was classified as severe in 43 patients (78.2%), while 12 (21.8%) had mild or moderate disease at admission. Comorbidities were prevalent among the deceased patients: chronic obstructive pulmonary disease (COPD) was present in 11 patients (20.0%), while diabetes mellitus and chronic kidney disease were each present in 6 (10.9%) and 4 (7.3%) patients, respectively. Cardiovascular disease (CVD) was the most prevalent comorbidity (27 patients; 49.1%), and 9 patients (16.4%) had active cancer. The median number of coexisting conditions was 2 (IQR: 1–3). Specifically, 17 patients (32.7%) had one comorbidity, while 28 (50.9%) had two or more. The median age of the deceased during the pre-Omicron period was 77 years (IQR: 67–86), 37 (97.4%) had severe disease at baseline while 8 (21.1%) had one and 26 (68.4%) had two or more comorbidities.

Regarding vaccination status, 40 of the deceased patients (72.7%) were vaccinated at least once, with 12 of them (21.8%) having received a full vaccination series, 24 (43.6%) having additionally received one booster, and 4 (7.3%) two boosters. Fifteen patients (27.3%) were unvaccinated. The median time since the last vaccination was 9 months (IQR: 6–12 months).

### 3.4. Effectiveness of Remdesivir

The estimated probability of discharge within 15 days was similar in the two periods (Omicron period 80.7%, 95% CI: 77.5–83.7; Pre-Omicron period 80.9%, 95% CI: 78.4–83.3). However, by day 30, the probability of discharge was lower in the Omicron period (88.1%, 95% CI: 85.4–90.5) compared to the Pre-Omicron period (93.0%, 95% CI: 91.3–94.5; *p* = 0.002) (Figure 3A).

When stratifying by disease severity (Figure 3B), among mild/moderate cases, Omicron patients had a significantly shorter median time to discharge (5 days, 95% CI: 5–6) compared to Pre-Omicron patients (9 days, 95% CI: 8–10; *p* < 0.001) while in severe cases the difference was lower (Omicron: 9 days; 95% CI: 8–10, Pre-Omicron: 10 days; 95% CI: 10–11) and not statistically significant (*p* = 0.396). Stratification by age (Figure 3C) showed significant (*p* < 0.001) differences for both below and above 65 years old cases with those in the Omicron period having shorter median (95% CI) times (5; 5–6 and 8; 7–8, respectively) to discharge compared to those in the Pre-Omicron period (9; 8–9 and 11; 11–12, respectively).

In multivariable analysis (Table 2), adjusting for age, Charlson Comorbidity Index (without age contribution), timing of remdesivir initiation and region of residence, patients hospitalized during the Omicron period, with mild or moderate disease had a significantly higher probability of discharge within 30 days compared to those in the Pre-Omicron period (adjusted Hazard Ratio-aHR: 1.91, 95% CI: 1.57–2.34; *p* < 0.001). The corresponding aHR in patients with severe disease at baseline was 1.11 (95% CI: 0.96–1.28; *p* = 0.176) suggesting that probabilities of discharge in the two periods were similar in severe cases (interaction between period and disease severity *p*-value < 0.001).

Older age was a strong predictor of prolonged hospitalization, with the probability of discharge decreasing significantly in patients aged 60–69 years (aHR: 0.72, 95% CI: 0.61–0.85, *p* < 0.001), 70–79 years (aHR: 0.65, 95% CI: 0.55–0.78, *p* < 0.001), and 80+ years (aHR: 0.48, 95% CI: 0.40–0.57, *p* < 0.001) compared to those aged 19–49 years. Comorbidities were also associated with lower probabilities of discharge (aHR: 0.95, 95% CI: 0.90–1.00, *p* = 0.047 per 1 unit increase in Charlson Comorbidity Index without the age contribution). Patients residing in rural areas had also decreased probabilities of discharge (aHR: 0.70, 95% CI: 0.55–0.88, *p* = 0.003) compared to those in urban or semi-urban areas. Finally, those initiating remdesivir at the same or next day after admission had higher probabilities of discharge compared to those with delayed (2 days or more after admission) remdesivir initiation (aHR: 1.32, 95% CI: 1.16–1.52, *p* < 0.001).

In the sample of Omicron period patients, prior full vaccination or previous COVID-19 infection within the past 6 months (n = 94; 15.5%) was not significantly associated with discharge time in both univariable (*p* = 0.649) or multivariable (*p* = 0.630) analyses. However, prior full vaccination at any time (n = 408; 67.3%) was associated with higher probabilities of discharge (aHR: 1.26, 95% CI: 1.05–1.53, *p* = 0.014; adjusted for all factors presented in Table 2).

### 3.5. Side Effects

Abnormal liver tests were significantly less frequent during the Omicron period compared to the Pre-Omicron period (1.7% vs. 12.9%, *p* < 0.001). Among patients hospitalized during the Omicron period, 5 patients (0.8%) had transaminase levels elevated 3–5 times the upper normal limits (ULN), and 4 patients (0.7%) had elevations exceeding >5 times ULN. In contrast, during the Pre-Omicron period, 9.3% of patients exhibited transaminase elevations of 3–5 times ULN, and 3.7% exceeded >5 times ULN. The median duration of transaminase elevations during the Omicron period was 2 days (IQR: 1–4) for 3–5 times ULN and 4.5 days (IQR: 2.5–5.5) for >5 times ULN, compared to longer durations reported during the Pre-Omicron period (median 3; IQR 2–5 and median 7; IQR 6–8 days, respectively).

During the Omicron period, abnormal transaminase levels were observed before remdesivir administration in 3 patients (0.5%), during treatment in 6 patients (1.0%), and after treatment completion in 1 patient (0.2%). In comparison, during the Pre-Omicron period, transaminase abnormalities occurred before remdesivir administration in 3.9% of patients, during treatment in 5.1%, and after treatment in 0.3%.

Acute kidney injury (AKI) was also rare during the Omicron period, affecting 6 patients (1.0%). Of these, 4 patients (0.7%) had AKI stage 1, and 1 patient each had AKI stages 2 and 3 (0.2%). The median duration of AKI was 3 days (IQR: 2.0–4.5), with 5 cases (0.8%) observed after remdesivir cessation. No AKI cases were reported during or within two days after remdesivir administration. These findings were consistent with the Pre-Omicron period, where AKI was similarly uncommon.

## 4. Discussion

In this study, a retrospective analysis from 1610 hospitalized patients who received remdesivir as part of their COVID-19 care during two distinct periods of the pandemic in Greece, corresponding to the dominance of the Pre-Omicron and Omicron variants, was performed. During the Omicron period, patients were hospitalized, with a notably older median age (median 76 vs. 61 years) and higher prevalence of comorbidities compared to the Pre-Omicron period. The median time from admission to remdesivir administration was also significantly shorter in the Omicron period than in the Pre-Omicron period. Most patients received the standard 5-day remdesivir regimen, with only 6.6% receiving remdesivir for 6–10 days during the Omicron period with the majority of them having active cancer and/or cardiovascular disease. Stratifying by baseline disease severity and adjusting for age and other risk factors, the Omicron period was characterized by shorter discharge times compared to the Pre-Omicron period but only in patients with mild/moderate disease at baseline. Early administration of remdesivir was also associated with faster discharge. These findings highlight the evolving clinical profile of hospitalized COVID-19 patients and the consistent effectiveness of remdesivir across variant periods.

Most patients were treated with a 5-day regimen regardless of the dominant variant. As for the Omicron period, 3-day regimens were observed among patients with an optimal clinical status and early discharge and longer regimens for patients with multiple comorbidities or immunosuppression [37,38].

The findings of our study validate the continued effectiveness of remdesivir in mitigating severe outcomes, even in the context of the highly transmissible Omicron variant. Early administration of remdesivir, something commonly observed in Omicron era with most patients receiving the first dose of remdesivir immediately after admission, continues to result in early optimal outcomes. Significant clinical improvement was observed among patients hospitalized during the Omicron period, as evidenced by the progressive shift in ordinal scale categories over time with 42.7% discharged by day 7 and 72.3% by day 14, mirroring trends seen in earlier variant waves [26,39]. This supports the strategy of prompt antiviral intervention to inhibit viral replication during its peak, a principle also established in other respiratory tract infections, such as influenza treatment [40].

The mortality rate of 9.1% observed by day 30 reflects the older and more comorbid profile of hospitalized patients during the Omicron period, underscoring COVID-19 as a continuing health challenge in vulnerable populations. However, this is consistent with global trends and emphasizes the need for early therapeutic interventions [41]. Although crude mortality during the Omicron period appeared higher compared to the Pre-Omicron period, this difference was primarily explained by the older age and higher comorbidity burden of patients admitted during the Omicron wave and the variant period itself seems not to be directly associated with mortality. Additionally, according to physicians the vast majority of deaths in the Omicron era (50/55) were not attributed to COVID-19 while 31/38 deaths in the Pre-Omicron period were attributed to COVID-19.

Although the Omicron variant is characterized by higher transmissibility and lower pathogenicity compared to previous variants, studies have shown that excess mortality during the Omicron period can remain significant, particularly in populations with low prior exposure to the virus or insufficient vaccination coverage [42]. In our cohort, recent vaccination or prior infection was not significantly associated with time to discharge noting though that only a relatively small proportion of patients (15.5%) had such recent immunological protection. In contrast, individuals with a history of full vaccination at any point had a higher likelihood of earlier discharge and lower mortality risk, supporting a sustained protective effect.

In addition, previous poor overall health status as well as socioeconomic status have been attributed to excess non-COVID-19 mortality for hospitalized patients in the Omicron era [42,43]. This is consistent with our findings, where most deceased patients during the Omicron period were elderly with multiple comorbidities, indicating that frailty and underlying disease rather than COVID-19 itself largely explained the observed deaths. International reports further emphasize that differences in excess deaths between Omicron and earlier waves may partly reflect evolving practices in attributing deaths ‘with’ versus ‘from’ COVID-19 [44,45,46]. The aforementioned findings are in line with recent reports indicating that remdesivir remains effective in reducing severe outcomes across different SARS-CoV-2 variant periods, including Pre-Omicron and Omicron [26,30,47,48]. Dobrowolska et al. demonstrated that remdesivir significantly reduced mortality during both the Pre-Omicron wave (OR = 0.42, 95% CI: 0.29–0.60; *p* < 0.0001) and the Omicron-dominated period (OR = 0.56, 95% CI: 0.35–0.92; *p* = 0.02), with 86.8% of Omicron-era patients receiving remdesivir within the first 5 days compared to 57.2% in the Pre-Omicron era [26]. In the same line, Chokkalingam et al., Pitts et al., and Takashita et al. confirmed the maintained efficacy of remdesivir across variant periods, supporting its continued clinical utility [26,30,47,48]. In line with these international findings, our study—a large real-world analysis in Greece including more than 1600 hospitalized patients—adds further evidence by showing that remdesivir maintained its effectiveness across both periods, despite the markedly older age and higher comorbidity burden of Omicron-era patients. Importantly, we demonstrated that early administration of remdesivir was consistently associated with shorter discharge times, particularly among patients with mild or moderate disease at baseline. This underscores the clinical relevance of prompt antiviral therapy, and is consistent with other studies highlighting that earlier initiation of remdesivir contributes to improved clinical outcomes [26,27].

ReEs-COVID19 revealed a significantly lower incidence of abnormal liver tests during the Omicron period compared to Pre-Omicron, with fewer patients experiencing elevated transaminase levels along with shorter durations of abnormalities. AKI was rare in both periods, with no cases occurring during or immediately after remdesivir administration in Omicron era. It should be noted that SARS-CoV-2 infection itself can cause liver toxicity, and emerging evidence suggests that variants differ in the degree of liver involvement [49]. A systematic review and meta-analysis by Chen et al. (2023) found that remdesivir use did not cause liver damage, with a relative risk (RR) of 0.87 (95% CI: 0.68–1.11), indicating no significant increase in hepatic adverse events compared to control treatments [50]. In the REDPINE study, remdesivir was generally well tolerated in patients with moderate to severe renal impairment hospitalized for COVID-19 pneumonia without liver abnormalities [51]. Additionally, a nationwide retrospective study in Italy by Pieralli et al. (2023) reported that remdesivir was well tolerated, with adverse drug reactions (ADRs) occurring in only 2.3% of patients, further supporting its hepatic safety profile [52].

The REDPINE study also demonstrated that remdesivir can be administered in patients with any stage of renal disease, including those on dialysis, without dosage adjustment. No new adverse reactions were identified, and adverse events (all grades) were reported in 8% of patients in the remdesivir group, comparable to the placebo group [51]. Furthermore, Burhan et al. (2023) in Indonesia found no significant differences in safety endpoints, including renal function, between patients receiving remdesivir and those who did not, suggesting that remdesivir does not exacerbate renal impairment in COVID-19 patients [53]. The safe profile of remdesivir is also proved for pediatric patients by a Phase-2/3, open-label trial, evaluated remdesivir in hospitalized children aged 28 days to 17 years with confirmed SARS-CoV-2 infection [54].

While the study provides valuable real-world insights, the retrospective and single-arm design limits the ability to perform direct comparisons with other therapeutic options. Furthermore, during the study period, some patients were excluded due to their enrollment in clinical trials, which may have led to a selection bias favoring less severe cases. However, this is unlikely to have biased our findings, as these trials included individuals across the full spectrum of disease severity and the number of excluded patients was small. Data collection from a single country healthcare network might also limit the generalizability of findings to other populations and healthcare settings. Another limitation of our study is that physicians had more treatment options and clinical experience during the Omicron period, which may have influenced patient outcomes independently of variant differences. The lower use of dexamethasone and anticoagulants during the Omicron period observed in our study reflects updated guideline recommendations rather than selective treatment of sicker patients, and is unlikely to have meaningfully affected outcomes. Finally, the absence of longer-term follow-up prevents the assessment of potential late complications or sustained efficacy of remdesivir and the absence of a control group limits the ability to directly attribute the observed clinical benefits to remdesivir.

Despite its limitations, this study has notable strengths. It provides a comprehensive evaluation of remdesivir use across two distinct SARS-CoV-2 variant periods, capturing valuable real-world data during the Pre-Omicron and Omicron waves. The inclusion of data from a diverse range of healthcare settings, spanning urban, semi-urban, and rural hospitals, strengthens its applicability to various clinical contexts. Moreover, the large sample size enhances the robustness of the findings, allowing for detailed subgroup analyses, including safety assessments related to hepatic and renal functions. The focus on early administration and its impact on clinical outcomes highlights actionable insights for optimizing remdesivir use in practice.

Future research should focus on the effectiveness and safety of remdesivir when administered to subgroups of patients with specific characteristics (e.g., immunocompromised individuals due to organ transplantation or cancer, elderly patients with limited vaccination response) or in combination with other antiviral or immunomodulatory agents as it may offer synergistic benefits.

## 5. Conclusions

ReEs-COVID19 confirmed the effectiveness and safety of remdesivir across both Pre-Omicron and Omicron periods, even among older and more comorbid patients. Early initiation of treatment was consistently associated with faster recovery and discharge, highlighting the importance of prompt antiviral therapy in hospitalized COVID-19 patients. Future studies should further explore its role in vulnerable subgroups, such as immunocompromised individuals, and in combination strategies with other therapeutic agents.

## Figures and Tables

**Figure 1 microorganisms-13-02242-f001:**
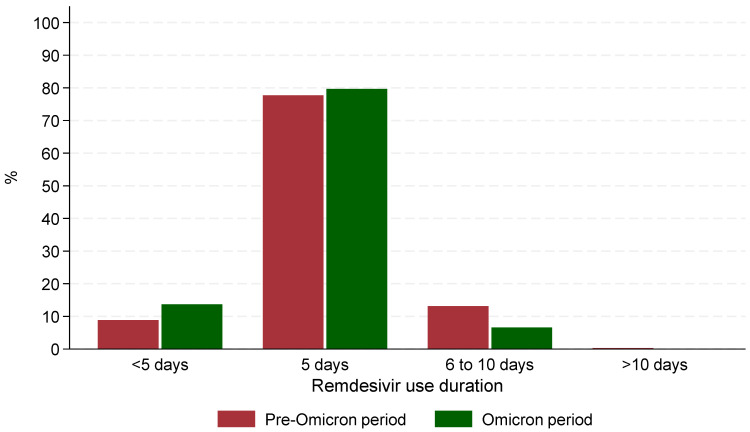
Patterns of remdesivir use during the 2 periods of the study.

**Figure 2 microorganisms-13-02242-f002:**
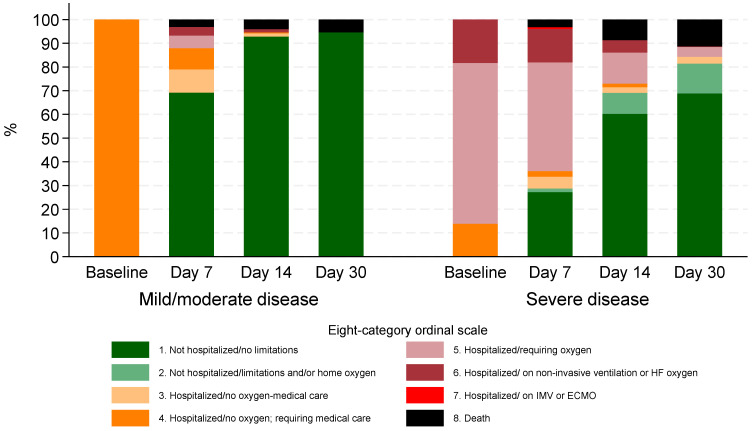
Clinical status according to eight-category ordinal scale for patients hospitalized during Omicron period by disease severity at baseline.

**Figure 3 microorganisms-13-02242-f003:**
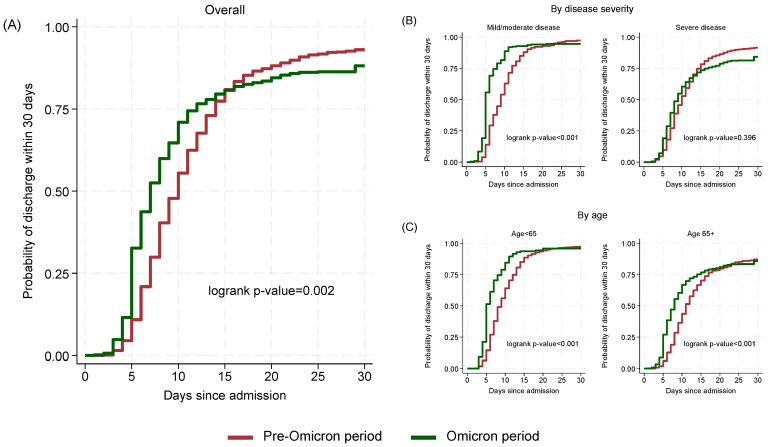
Cumulative probability of discharge within 30 days by time since admission and period: overall (**A**), by disease severity (**B**) and by age (**C**).

**Table 1 microorganisms-13-02242-t001:** Patients’ characteristics, coexisting conditions and laboratory markers by period of hospitalization at baseline (i.e., hospital admission).

Variable	Pre-Omicron Variant n = 1004 (62.4%)	Omicron Variant n = 606 (37.6%)	Overall n = 1610 (100%)	*p*-Value ^1^
**Demographics**				
Sex, male	608 (60.6%)	319 (52.6%)	927 (57.6%)	0.002
Age (years)	61 (51, 73)	76 (66, 85)	67 (54, 78)	<0.001
Age (in groups; years)				<0.001
*– 19–49*	215 (21.4%)	52 (8.6%)	267 (16.6%)	
*– 50–59*	237 (23.6%)	45 (7.4%)	282 (17.5%)	
*– 60–69*	244 (24.3%)	99 (16.3%)	343 (21.3%)	
*– 70–79*	186 (18.5%)	166 (27.4%)	352 (21.9%)	
*– 80+*	122 (12.2%)	244 (40.3%)	366 (22.7%)	
Area of residence				<0.001
*– Rural area*	74 (7.4%)	15 (2.5%)	89 (5.5%)	
*– Semi-urban area*	830 (82.7%)	27 (4.5%)	857 (53.2%)	
*– Urban area*	100 (10.0%)	564 (93.1%)	664 (41.2%)	
Region of residence				<0.001
*– Rest of Greece*	578 (57.6%)	269 (44.4%)	847 (52.6%)	
*– Attika*	426 (42.4%)	337 (55.6%)	763 (47.4%)	
Greek nationality	919 (91.5%)	576 (95.0%)	1495 (92.9%)	0.009
**Baseline severity**				
Disease severity				<0.001
*– Severe* ^2^	745 (74.2%)	382 (63.0%)	1127 (70.0%)	
*– Mild/moderate*	259 (25.8%)	224 (37.0%)	483 (30.0%)	
Baseline oxygen need				0.001
*– Not requiring O_2_*	394 (39.2%)	277 (45.7%)	671 (41.7%)	
*– Requiring low flow O_2_*	524 (52.2%)	259 (42.7%)	783 (48.6%)	
*– Requiring high flow O_2_ or mechanical ventilation*	86 (8.6%)	70 (11.6%)	156 (9.7%)	
**Comorbidities and scores**				
Chronic Obstructive Pulmonary Disease	62 (6.2%)	93 (15.3%)	155 (9.6%)	<0.001
Diabetes mellitus	236 (23.5%)	165 (27.2%)	401 (24.9%)	0.096
Cardiovascular disease	101 (10.1%)	354 (58.4%)	455 (28.3%)	<0.001
Coronary artery disease	91 (9.1%)	85 (14.0%)	176 (10.9%)	0.002
Immunosuppression	27 (2.7%)	30 (5.0%)	57 (3.5%)	0.025
Cancer	49 (4.9%)	107 (17.7%)	156 (9.7%)	<0.001
Chronic kidney disease	17 (1.7%)	36 (5.9%)	53 (3.3%)	<0.001
Chronic liver disease	10 (1.0%)	1 (0.2%)	11 (0.7%)	0.061
Number of comorbidities				<0.001
*– None*	292 (29.1%)	88 (14.5%)	380 (23.6%)	
*– One*	229 (22.8%)	125 (20.6%)	354 (22.0%)	
*– Two or more*	483 (48.1%)	393 (64.9%)	876 (54.4%)	
Charlson Comorbidity Index	2 (1, 4)	4 (3, 5)	3 (1, 4)	<0.001
4C Mortality Score ^3^	7 (5, 9)	10 (7, 12)	8 (5, 11)	<0.001
**Life-style factors**				
Alcohol abuse	9 (0.9%)	10 (1.7%)	19 (1.2%)	0.233
Smoking status				<0.001
*– Unknown*	309 (30.8%)	281 (46.4%)	590 (36.6%)	
*– Active smoker*	68 (6.8%)	64 (10.6%)	132 (8.2%)	
*– Never smoker*	455 (45.3%)	154 (25.4%)	609 (37.8%)	
*– Ex-smoker*	172 (17.1%)	107 (17.7%)	279 (17.3%)	

^1^: *p*-values refer to comparisons between Pre-Omicron and Omicron groups. Continuous variables were compared using the Mann–Whitney U test, and categorical variables using the Chi-square test (or Fisher’s exact test as appropriate). ^2^: In Pre-Omicron and Omicron periods 135/745 and 53/382 severe cases, respectively, did not require supplemental O_2_ but had either low O_2_ saturation (≤94%; 122 in Pre-Omicron and 51 in Omicron) or high respiratory rate (≥24/min; 13 in Pre-Omicron and 2 in Omicron). ^3^: 4C Mortality Score calculation does not include Glasgow Coma Scale (i.e., assuming normal consciousness for all).

**Table 2 microorganisms-13-02242-t002:** Multivariable Cox model for the probability of discharge within 30 days by variant period. Model is stratified by disease severity at baseline and includes interaction between variant and disease severity at baseline.

Covariate	Haz. Ratio	95% C.I.	*p*-Value ^1^
Variant period			
– *Omicron vs. Pre-Omicron (mild or moderate disease* ^2^*)*	1.91	(1.57, 2.34)	<0.001
– *Omicron vs. Pre-Omicron (severe disease* ^2^*)*	1.11	(0.96, 1.28)	0.176
Age (years)			
*– 50–59 vs. 19–49*	0.92	(0.78, 1.10)	0.370
*– 60–69 vs. 19–49*	0.72	(0.61, 0.85)	<0.001
*– 70–79 vs. 19–49*	0.65	(0.55, 0.78)	<0.001
*– 80+ vs. 19–49*	0.48	(0.40, 0.57)	<0.001
Charlson Comorbidity Index ^3^ (per unit)	0.95	(0.90, 1.00)	0.047
Remdesivir initiation			
*– 0–1 vs. 2+ days from admission*	1.32	(1.16, 1.52)	<0.001
Area			
*– Rural vs. Urban or semi-urban*	0.70	(0.55, 0.88)	0.003

^1^: *p*-values were derived from Wald-type tests in the multivariable Cox model. ^2^: Interaction between period and disease severity *p*-value < 0.001. ^3^: Calculated without the age contribution.

## Data Availability

ReEs-COVID 19 are derived from collaborating clinics and although individual data do not include patient names or identifying information of the participants, as data contain potentially sensitive information, there are ethical restrictions imposed by the Bioethics and Deontology Committee of the Medical School of the National and Kapodistrian University of Athens. Anonymized individual data can be shared after interested researchers submit a concept sheet to the AMACS steering committee (chair: Giota Touloumi, email: gtouloum@med.uoa.gr) and the Bioethics and Deontology Committee of the Medical School of the National and Kapodistrian University of Athens (bioethics@med.uoa.gr).

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
