# Peer review of "Remdesivir: Effectiveness and Safety in Hospitalized COVID-19 Patients—Analysis of Retrospectively Collected Data from Daily Practice in the Omicron Variant Era and Comparison with the Pre-Omicron Period"

_microorganisms, 2025, doi:10.3390/microorganisms13102242_

Round 1

Reviewer 1 Report

Comments and Suggestions for Authors

This manuscript addresses a relevant and timely topic: the effectiveness and safety of remdesivir in hospitalized patients with COVID-19, comparing the pre-Omicron and Omicron eras. The sample size is substantial (n=1,610), with data from multiple centers, which strengthens the findings and enables meaningful stratified analyses. The study provides valuable real-world evidence, an essential complement to clinical trial data.

However, there are areas where the presentation and analysis could be improved to enhance clarity, the robustness of conclusions, and the interpretation of results. Minor structural and editorial issues should also be addressed before publication:

  • Include a multivariate analysis of mortality adjusted for age, comorbidities, severity, and vaccination to avoid attributing differences to RDV without controlling for confounders.
  • Explain whether the exclusion of patients participating in clinical trials may have biased the sample toward less severe cases.
  • Clearly define “death attributable to COVID-19” and specify whether a consensus or standardized protocol was applied across hospitals.
  • Justify the assumption of a Glasgow Coma Scale score of 15 for all patients and discuss its potential impact on calculating the 4C Mortality Score.
  • If feasible, analyze whether the lower use of dexamethasone and anticoagulants during the Omicron era could have influenced outcomes, and explain how this factor was accounted for.
  • Qualify the conclusion regarding “higher mortality in Omicron” by clarifying that this is primarily due to older age and greater comorbidity burden, and present adjusted results.
  • Present COVID-19–attributable and non-attributable mortality separately, along with associated factors.
  • Improve the readability of Figures 1 and 3 by using more contrasting colors, consistent scales, and clearer legends.
  • In Table 1, group clinically relevant p-values and avoid visual overload with comparisons of limited clinical importance.
  • Streamline the discussion by reducing the extensive review of previous literature and focusing more on how the current findings confirm, refine, or contradict existing evidence.
  • Emphasize that the absence of a control group limits the ability to attribute benefits directly to RDV.

Author Response

We would like to thank the Reviewers for giving us the opportunity to submit a revised version of our manuscript. We appreciate the Reviewers’ insight and we hope that we accommodated their suggestions in the revised manuscript. All changes have been tracked in the revised manuscript.

REVIEWER #1

This manuscript addresses a relevant and timely topic: the effectiveness and safety of remdesivir in hospitalized patients with COVID-19, comparing the pre-Omicron and Omicron eras. The sample size is substantial (n=1,610), with data from multiple centers, which strengthens the findings and enables meaningful stratified analyses. The study provides valuable real-world evidence, an essential complement to clinical trial data.

However, there are areas where the presentation and analysis could be improved to enhance clarity, the robustness of conclusions, and the interpretation of results. Minor structural and editorial issues should also be addressed before publication:

Comment 1

Include a multivariate analysis of mortality adjusted for age, comorbidities, severity, and vaccination to avoid attributing differences to RDV without controlling for confounders.

Reply

We thank the Reviewer for this helpful suggestion. In the original manuscript, mortality was presented descriptively because our primary endpoint was discharge within 30 days, and all enrolled patients received remdesivir by design; therefore, differences cannot be attributed to treatment allocation. We agree that adjusted analyses are useful to clarify whether the observed mortality difference between the two periods is explained by differences in patient characteristics.

We have now fitted logistic regression models of 30-days mortality adjusting for age, comorbidities, and baseline severity. Vaccination was not included in the full-cohort model because no patients were vaccinated in the pre-Omicron period, making vaccination perfectly collinear with calendar period and therefore non-estimable. Instead, we (i) report a univariable and a multivariable model for the full cohort excluding vaccination, and (ii) provide a sensitivity analysis restricted to the Omicron period where vaccination varies and could be included (Supplementary Tables S2, S3 and S4). Results are reported (lines 255-260) and discussed (lines 385-389 and line 399) in the revised manuscript.

Comment 2

Explain whether the exclusion of patients participating in clinical trials may have biased the sample toward less severe cases.

Reply

We thank the Reviewer for this comment. Patients enrolled in clinical trials were excluded to avoid overlapping interventions and potential confounding. We do not believe this exclusion biased our cohort toward less severe cases. Clinical trials conducted during the study period were not limited to critically ill patients but also included studies of monoclonal antibodies and immunomodulators that enrolled patients across a wide range of disease severity. In addition, the number of excluded patients was small relative to the overall cohort, and thus this criterion is unlikely to have affected the representativeness or size of our study sample. We clarify this issue in our methods and discussion sections (lines 109-111 and lines 455-457)

Comment 3

Clearly define “death attributable to COVID-19” and specify whether a consensus or standardized protocol was applied across hospitals.

Reply

We appreciate the Reviewer’s suggestion. In our study, the classification of a death as “attributable to COVID-19” was based on the clinical judgment of the treating physicians, who documented the primary cause of death in the medical records. This practice was part of routine hospital documentation and followed the national reporting standards. For this reason, we consider the attribution reliable, even though there was no centralized adjudication committee. This clarification has been added in the Results section immediately after the description of COVID-19–attributable deaths (lines 253-254).

Comment 4

Justify the assumption of a Glasgow Coma Scale score of 15 for all patients and discuss its potential impact on calculating the 4C Mortality Score.

Reply

We thank the Reviewer for raising this important point. In this retrospective dataset, GCS at admission was not routinely recorded, so we assumed GCS = 15 when calculating the 4C Mortality Score. We acknowledge that this may have led to a slight underestimation of the score in a minority of patients with impaired consciousness. However, (i) patients directly admitted to ICU were excluded, (ii) markers of acute illness such as baseline severity and oxygen requirement were explicitly captured and adjusted for, and (iii) the 4C score was used only for descriptive characterization and not in our main analyses. We have clarified this assumption in the Methods (lines 119-120).

Comment 5

If feasible, analyze whether the lower use of dexamethasone and anticoagulants during the Omicron era could have influenced outcomes, and explain how this factor was accounted for.

Reply

We thank the Reviewer for this comment. As noted in the manuscript, the use of dexamethasone and anticoagulants was lower during the Omicron period. This difference reflects changes in national and international guidelines rather than selective treatment, as dexamethasone was no longer recommended for all hospitalized patients and anticoagulant use was adapted according to updated guidance. While treatment-by-indication bias can occur in observational studies, in this case the differences in therapy largely mirror evolving standards of care, and we do not believe they meaningfully influenced outcomes. We briefly refer to this issue in the Discussion of the updated manuscript (lines 461-464).

Comment 6

Qualify the conclusion regarding “higher mortality in Omicron” by clarifying that this is primarily due to older age and greater comorbidity burden, and present adjusted results.

Reply

We thank the Reviewer for this valuable suggestion. As recommended, we have qualified our conclusions regarding mortality during the Omicron period. Specifically, in the revised Discussion (lines 385-389), we now emphasize that the higher crude mortality observed during the Omicron period was mainly explained by the older age and higher comorbidity burden of these patients, rather than by the variant itself.

Comment 7

Present COVID-19–attributable and non-attributable mortality separately, along with associated factors.

Reply

We thank the Reviewer for this thoughtful suggestion. We agree that distinguishing between COVID-19–attributable and non-attributable mortality is clinically relevant. In the revised manuscript (Results section), we present COVID-19–attributable and non-attributable deaths separately for descriptive purposes. However, given the limited number of events in each subgroup, additional regression analyses exploring associated factors would be underpowered and potentially misleading. Furthermore, mortality was not the primary outcome of our study, which was designed to assess time to discharge, patterns of RDV use and its safety profile. For these reasons, we believe that further analyses would not add reliable or meaningful insights, but we highlight the observed differences descriptively.

Comment 8

Improve the readability of Figures 1 and 3 by using more contrasting colors, consistent scales, and clearer legends.

Reply

We thank the reviewer for their comment regarding the readability of Figures 1 and 3. The figures use high-contrast colors, consistent scales across subpanels, and explanatory legends. They are presented in color, which we believe aids interpretation, but we would be happy to make further improvements and would appreciate any specific guidance regarding which aspects the reviewer would like clarified or emphasized.

Comment 9

In Table 1, group clinically relevant p-values and avoid visual overload with comparisons of limited clinical importance.

Reply

We thank the reviewer for this valuable comment. We have streamlined Table 1 to improve readability by (i) presenting only the “Yes” category for binary variables, (ii) removing some abbreviations, and (iii) rearranging the characteristics into clinically meaningful groups (demographics, baseline severity, comorbidities, etc.).

Comment 10

Streamline the discussion by reducing the extensive review of previous literature and focusing more on how the current findings confirm, refine, or contradict existing evidence.

Reply

We thank the Reviewer for this constructive comment. We have taken this into account and revised the Discussion in the sections where the reference to previous literature was predominant, to make the text more concise and to highlight more clearly the specific contribution of our study.

 Comment 11

Emphasize that the absence of a control group limits the ability to attribute benefits directly to RDV.

Reply

We thank the Reviewer for this important comment. We have now explicitly acknowledged in the Limitations section that the absence of a control group restricts our ability to directly attribute the observed benefits to remdesivir. This clarification has been added in the revised manuscript (lines 465-467)

Reviewer 2 Report

Comments and Suggestions for Authors

The article submitted by Nikos Pantazis et al to Microorganisms is devoted to a retrospective study of the effect of the Remdesivir on the COVID-19

Some issues should be resolved prior the paper can be recommended for publication

1) explain ReEs-COVID19 in the lines 36 and 97

2) change the keywords: add Remdesivir, add Omicron;

3) replace the abbreviation RDV with Remdesivir, so the article will be more understandable to the readers and probably will be better indexed by search engines. There is little space saving from using this abbreviation.

4) Explain in Table 1 what does p-values ​​mean, and which data are compared (lines, columns), which statistical criterion was used, and why. The same is relevant to Table 2.

5) Add to the chapter 2.4 information regarding the use of particular statistical criteria

6) I believe that the Introduction section should include a formulation of the hypothesis that the authors tested during the study. Also, it would be useful for the reader if a Conclusion section is added to the article, where the authors describe in 2-3 sentences the conclusions they draw from the results obtained, and also indicate whether the hypothesis was proven (or disproved).

Author Response

We would like to thank the Reviewers for giving us the opportunity to submit a revised version of our manuscript. We appreciate the Reviewers’ insight and we hope that we accommodated their suggestions in the revised manuscript. All changes have been tracked in the revised manuscript.

REVIEWER #2

The article submitted by Nikos Pantazis et al to Microorganisms is devoted to a retrospective study of the effect of the Remdesivir on the COVID-19

Some issues should be resolved prior the paper can be recommended for publication

Comment 1

Explain ReEs-COVID19 in the lines 36 and 97

Reply

We thank the Reviewer for this helpful suggestion. As recommended, we clarified the acronym by adding the full study title “ReEs-COVID19 (Remdesivir Effectiveness and Safety in COVID-19)” at its first mention in full text (line 88). The acronym is then used consistently in the rest of the manuscript as well as in the abstract.

Comment 2

Change the keywords: add Remdesivir, add Omicron;

Reply

Keywords have been updated accordingly.

Comment 3

Replace the abbreviation RDV with Remdesivir, so the article will be more understandable to the readers and probably will be better indexed by search engines. There is little space saving from using this abbreviation.

Reply

We thank the Reviewer for this comment. All occurrences of RDV have been replaced with “remdesivir” in the revised manuscript.

Comment 4

Explain in Table 1 what does p-values ​​mean, and which data are compared (lines, columns), which statistical criterion was used, and why. The same is relevant to Table 2.

Reply

In Table 1, we have added a footnote clarifying that p-values refer to comparisons between Pre-Omicron and Omicron groups, with categorical variables compared using Chi-square tests (or Fisher’s exact test as appropriate) and continuous variables with Mann-Whitney. For Table 2, we revised the footnote to specify that p-values are derived from Wald-type tests in the multivariable Cox proportional hazards model.

Comment 5

Add to the chapter 2.4 information regarding the use of particular statistical criteria

Reply

In the revised version (lines 156-162), we clarify that categorical variables were compared using Chi-square tests (or Fisher’s exact test as appropriate), continuous variables using Mann-Whitney, and time-to-event outcomes with Kaplan-Meier curves, log-rank tests, and Cox proportional hazards models. A p-value <0.05 was considered statistically significant.

Comment 6

I believe that the Introduction section should include a formulation of the hypothesis that the authors tested during the study. Also, it would be useful for the reader if a Conclusion section is added to the article, where the authors describe in 2-3 sentences the conclusions they draw from the results obtained, and also indicate whether the hypothesis was proven (or disproved).

Reply

In the revised manuscript, we have added a clear formulation of our study hypothesis at the end of the Introduction (lines 91-94). Furthermore, we included a concise Conclusion section summarizing the main findings in 2–3 sentences and stating that our hypothesis was confirmed.

Round 2

Reviewer 1 Report

Comments and Suggestions for Authors

The article is ready for publication. Congratulations!

Reviewer 2 Report

Comments and Suggestions for Authors

The authors resolved the issues and the manuscript might be recommended for publication